# Relationships between Physical Activity, Sedentary Behaviour and Cognitive Functions in Office Workers

**DOI:** 10.3390/ijerph16234721

**Published:** 2019-11-27

**Authors:** Emil Bojsen-Møller, Carl-Johan Boraxbekk, Örjan Ekblom, Victoria Blom, Maria M. Ekblom

**Affiliations:** 1The Swedish School of Sport and Health Sciences, GIH, 11486 Stockholm, Sweden; orjan.ekblom@gih.se (Ö.E.); victoria.blom@gih.se (V.B.); maria.ekblom@gih.se (M.M.E.); 2Danish Research Center for Magnetic Resonance, Center for Functional and Diagnostic Imaging and Research, Copenhagen University Hospital, 2650 Hvidovre, Denmark; CJ@drcmr.dk; 3Institute of Sports Medicine Copenhagen, Copenhagen University Hospital Bispebjerg, 2400 Copenhagen, Denmark; 4Department of Radiation Sciences, Umeå University, 90187 Umeå, Sweden; 5Department of Neuroscience, Karolinska Institutet, 17177 Stockholm, Sweden

**Keywords:** executive functions, episodic memory, cognition, physical activity, sedentary behaviour, office workers

## Abstract

Increasing evidence from animal experiments suggests that physical activity (PA) promotes neuroplasticity and learning. For humans, most research on the relationship between PA, sedentary behaviour (SB), and cognitive function has relied on self-reported measures of behaviour. Office work is characterised by high durations of SB combined with high work demands. While previous studies have shown that fitter office workers outperform their less fit colleagues in cognitive tests, the importance of PA and SB remains unknown. This study investigated associations between objectively measured PA and SB, using hip-worn accelerometers, and cognitive functions in 334 office workers. Time spent in moderate-to-vigorous PA (MVPA) was not associated with any cognitive outcome. However, time spent in SB tended to be positively associated with words recalled in free recall (β = 0.125). For the least fit participants, the average length of MVPA bouts was favourably related to Stroop performance (β = −0.211), while for the fitter individuals, a longer average length of MVPA bouts was related to worse recognition (β = −0.216). While our findings indicate that the length of MVPA bouts was associated with better Stroop performance in the least fit participants, our findings do not support the notion that more time spent in MVPA or less time in SB is associated with better cognitive function.

## 1. Introduction

There is substantial evidence that being physically active reduces the risk for all-cause mortality, the incidence of cardio-metabolic diseases, and some types of cancer [1]. In addition, increasing evidence suggests that the health benefit of being physically active extends to the brain and many of its cognitive functions [2,3]. Evidence from animal studies suggests that physical activity (PA) promotes neuroplasticity—a foundation for cognitive function and learning [2]. PA can be defined as any bodily movement produced by skeletal muscles that increases energy expenditure above resting state [4], and is characterized by its modality, frequency, duration, and intensity [5]. The intensity of PA is often categorized into low (LIPA, including activities such as slow walking), moderate (MPA, e.g., fast walking), and vigorous (VPA, e.g., climbing stairs or running). Physical fitness or, in particular cardiorespiratory fitness (CRF), is often incorrectly used interchangeably with PA. For healthy adults, the majority of research on the relationship between PA, sedentary behaviour (SB), and cognitive function has relied on different proxy measures of behaviour.

Our group recently revealed that up to a maximal oxygen consumption (VO_2max_) of 44 mL⋅kg^−1^⋅min^−1^, there is a linear relationship between VO_2max_ and both inhibition and episodic recognition in a middle-aged working population [6]. However, it is unknown whether objective measures of PA and SB are related to cognitive functions and, if so, whether these associations are significant after taking the CRF into account. Although PA may be related to CRF in adults [7], PA reflects a more complex movement pattern that may affect cognitive function through multiple mechanisms [8].

While PA has been associated with cognitive function [9], the vast majority of studies rely on self-reported measures of habitual PA [10], which have been shown to have poorer validity compared to objective measures [11]. Furthermore, most studies have addressed either children, where the brain is still developing, or older individuals who are at higher risk for cognitive decline [10]. Limited evidence exists on the relationship between modes of objectively measured PA and cognitive functions in a working-age population.

After taking the effect of PA into consideration, SB has been shown to have adverse systemic health effects [12], but only for people that do not meet the global recommendations for PA [13]. This is particularly alarming in high-income countries, where employees are often in occupations characterized by extended periods of sitting and with limited or no demands for PA, such as office work [14]. SB has been defined as any waking hour behaviour with an energy expenditure of ≤1.5 metabolic equivalents while in a sitting or reclining posture [15]. Office workers are often sedentary for prolonged bouts and these extended periods have been associated with increased cardio-metabolic risk [16]. To date, limited evidence exists on how SB is related to cognitive function. In a recent systematic review on SB and cognition, Falck and colleagues reported finding only eight studies that met their inclusion criteria [17]. The review reported that SB was negatively associated with cognitive function. However, the only study using objective measures of SB did not find any associations between SB and cognitive function [18].

In summary, most research studies on the relationship between PA, SB, and cognitive function have relied on self-reported behaviour measures in children or older individuals. Thus, research relying on objective measures of PA and SB is needed for more accurate investigations of the relationship between different intensities and durations of PA, SB, and cognitive function in working-age adults. To properly address the difficulties in using a self-reported measurement of PA [11] and SB [19], we aimed to investigate whether and to what degree objective measures of PA and SB are related to cognitive functions among office workers. We hypothesized that more average daily time spent in moderate-to-vigorous PA (MVPA) and less time spent in SB would be related to better cognitive task performance independently of CRF and that the associations would be even stronger for prolonged bouts of PA and SB in less fit individuals.

## 2. Materials and Methods

Participants/study population (See Table 1 for demographics). In a cross-sectional design, 1940 office workers from two Swedish companies (Intrum, Solna, Sweden and ICA-gruppen, Stockholm, Sweden) were invited to participate in the study. These companies were part of a research collaboration to coproduce knowledge in a series of studies on how PA and SB relate to healthy brain function. Of the 1940 invited employees, 547 (28 %) answered the questionnaire and 369 (19%) participants took part in all measurements. A more detailed description of the sample is described elsewhere [6]. The study was approved by the Regional Ethical Review Board, Stockholm, Sweden (2016/796-31) and was carried out in accordance with the Helsinki Declaration II.

### 2.1. Procedure

Participants were asked to answer a detailed questionnaire about their lifestyle, work environment, and mental health. Additionally, participants were invited to attend a session containing cognitive testing, a submaximal aerobic fitness test [20], and instruction on activity monitoring. The session took place at the workplace of the participants. The duration was approximately one hour and 15 min, and participants were given permission to attend during working hours. The testing was carried out by test leaders specifically educated to perform cognitive and fitness testing.

After receiving verbal and written information about the study, participants gave their written informed consent. Firstly, participants received instruction and they then were equipped with activity monitors. Activity measurement was then obtained over the subsequent eight days. Secondly, cognitive testing was performed; and finally, a submaximal aerobic fitness test was completed. More details on the test session are described below.

### 2.2. Measures

#### 2.2.1. Activity Monitoring

Objective measures of daily movement patterns were obtained using a hip-worn triaxial accelerometer, Actigraph^®^ GT3X (Actigraph, Pensacola, FL, USA). Participants were instructed to wear the accelerometer for eight consecutive days on the right hip during waking hours, except during water-based activities.

The software ActiLife (version 6.13.3) (Actigraph, Pensacola, FL, USA) was used to initialize the accelerometry sampling procedure (30 Hz) and to subsequently download and process the data. The data were downloaded and transformed into an arbitrary unit named “counts”. The 30 Hz signal was downsampled and stored into epochs of 60 s [21]. PA intensity was then presented as counts per minute (cpm). Participants with at least four days containing at least 600 min of valid wear time were included in the analysis [22]. No more than two weekend days were allowed. Activity diaries were used to identify time in and out of bed. Sixty uninterrupted minutes of zero counts were classified as non-wear time [23]. Wear time was calculated by subtracting non-wear time and time in bed from 24 h.

In order to identify intensity level, we used the procedure according to Sasaki et al. 2011 [24] with 200 cpm as the maximal cut-off point for sedentary intensity [25]. A sedentary bout was defined as at least 20 consecutive minutes with no more than 199 cpm. MVPA was defined as activities eliciting at least 2690 cpm, and bouts of MVPA were defined as a minimum of 10 min consecutive activity with a least 2690 cpm. Drop time was allowed for maximal 2 min.

The outcomes derived from the Actigraph were percentage time spent in MVPA (% in MVPA; time in MVPA divided by total wear time) and in SB (% in SB; sedentary time divided by total wear time), and daily average minutes spent in MVPA bouts and in SB bouts. Furthermore, the average length of MVPA bouts and the average length of SB bouts were extracted.

#### 2.2.2. Neuropsychological Test Battery

Cognitive testing consisted of a battery of nine tests assessing different cognitive domains. Six tests were computerized and three were pen and paper tests. The battery was similar to Jonasson et al. 2017 [26] with slight adaptations made to the trail making test A and B. All tests showed good validity and reliability for measurement of the respective cognitive domains [26]. For the computerised tests, E-Prime 2.0 software (Psychology Software Tools, Pittsburgh, PA, USA) was used. The duration of the cognitive testing was approximately 45 min. All participants performed the cognitive tests in the following order: digit symbol, free recall, digit span backwards, n-back, word recognition a: encoding, automated operation span, Stroop, trail making test a, trail making test b, and word recognition b: retrieval. Preceding all tests, except free recall, a practice part was completed to ensure that the participant had understood the task instructions before continuing. Each test is described in the following section. For more details on the cognitive testing, see Pantzar et al. 2018 and Jonasson et al. 2017.

##### Episodic Memory

Free recall: A list of 16 nouns was presented to the subject visually. Immediately after the word presentation, participants were asked to recall as many words as possible and write them on a blank paper in any order. The number of correctly recalled words was the dependent outcome.

Word recognition: A list of 30 nouns was visually presented to the subject. The participants were required to encode the words and approximately 25 min later recall as many words as possible. During recollection, the subject indicated with a key press whether the word was a part of the list that was originally presented or not. The dependent outcome was number of correctly recalled words.

##### Processing Speed

Digit symbol: The subject identified with a key press whether a single digit–symbol combination could be identified on a line with nine digit–symbol combinations. The dependent measures were the number of correct answers and the response time on correctly answered trials expressed in seconds.

Trail making test A (TMT-A): Participants were instructed to connect circles containing numbers from 1 to 25, as fast as possible without lifting the pen from the paper. The dependent measure was the time it took to complete the task expressed in seconds.

##### Executive Functions

The 2–back (updating): Participants performed a computerised version of the n-back task. The objective was to indicate with a key press if the number shown was the one that was presented one stimulus (1–back) or two stimuli (2–back) earlier. The dependent outcome was 2–back accuracy and response time (RT) on correct answers.

Trail making test B (TMT-B) (shifting): The task was to connect circles as fast as possible without lifting the pen from the paper. Every other circle contained a number and every other contained a letter. The task was completed by switching between numbers and letters and connecting 1 with A, A with 2, 2 with B and so forth, until all connections were made. Participants were instructed to perform the test as fast as possible without making mistakes. The dependent measure was the time it took to complete the task expressed in seconds.

Stroop (inhibition): In order to test resistance to interference (inhibition), a modified version of the Stroop test was used. The participants were instructed to say the printed colour of the written word and not read the word. This was to be done as fast as possible without making any mistakes. The dependent outcome was the time it took to complete the task expressed in seconds.

##### Working Memory

Digit span backwards: A sequence of numbers was presented to be memorised in reverse order. For example, the correct answer to the sequence 3–5-9 was 9-5-3. Numbers were presented one at a time. Participants answered by pressing the correct number sequence on a keyboard. Sequences began with three numbers and continued until two incorrect answers were given on the same trial. The dependent outcome was the length of the longest correctly completed sequence.

Automated operation span (AOS): The objective of this task was to remember a sequence of letters while answering simple mathematical equations with true or false. The dependent measure was the sum of correctly remembered sets multiplied by the respective set size.

#### 2.2.3. Cardiorespiratory Fitness

The Ekblom–Bak submaximal fitness test was performed on a cycle ergometer (model 828E, Monark, Varberg, Sweden). Participants exercised on a standardised lower work rate (0.5 kilopond) for four minutes followed by an individualised higher work rate for four minutes. Maximal oxygen consumption (VO_2max_) was estimated from the change in heart rate divided by the change in work rate from the lower to the higher work rate, as described in more detail by Björkman et al. 2016 [20]. VO_2max_ was expressed as ml⋅kg^−1^⋅min^−1^. Body mass was measured using a standard scale. The Ekblom–Bak test has been shown to be valid and reliable with a high correlation coefficient to directly measured oxygen consumption (*r* = 0.90, for all participants) [20].

#### 2.2.4. Statistical Analysis

Statistical analysis was performed using IBM SPSS statistics version 24 (IBM, Armonk, NY, USA). Descriptive statistics were reported as the means ± standard deviations, unless stated otherwise. Normal distributions were tested using Shapiro–Wilk’s test of normality. Dependent and independent variables that did not meet the criterion for normality were logarithmic transformed to a base of 10. Multiple linear regression analysis was carried out using the cognitive outcomes as dependent measures and PA measurements as independent measures. Standardised beta values and 99% confidence intervals were reported. All multiple linear regression models were adjusted for age, gender, and education. All PA measurements were analysed both with and without adjustment for VO_2max_. Models containing bout measures of PA and SB were additionally adjusted for wear time. Additional regression analyses were performed in the two cognitive tests, Stroop and recognition, where we previously showed an association with VO_2max_ [6]. Fitness level was defined using the median of VO_2max_ and participants with a fitness level above and below the median were analysed separately. The threshold for statistical significance was set at *p* < 0.01, to adjust for multiple testing.

## 3. Results

Of the 369 subjects who participated, 337 had a sufficient amount of valid accelerometer data and three participants were excluded because of missing education data. Thus, 334 were included in the analysis. All included participants had valid accelerometer data on more than three working days. Descriptive data from the participating office workers are reported in Table 1.

Not all participants had valid data on every cognitive test. Therefore, the number of participants differs between the different cognitive tests (see Table 1).

### 3.1. Effects of Average Daily MVPA and SB

Multiple linear regressions with log % in MVPA and % in SB as independent variables are summarised in Table 2 and Table 3. All models were adjusted for age, gender, and education.

Upon controlling for known covariates, log % in MVPA was not significantly related to any of the cognitive outcomes, while % in SB tended to be positively related to better performance on free recall. Otherwise, % in SB was not related to any of the other cognitive outcomes. When VO_2max_ was added to the models % in SB still tended to be related to better performance on free recall, and % in MVPA remained unrelated to any cognitive outcome.

### 3.2. Effects of Bouts of MVPA and SB

Table 4 and Table 5 show the multiple regression models containing bout measures of PA. All models were adjusted for age, gender, and education. There was no relationship between daily average time spent in MVPA bouts and cognitive performance. The average length of the MVPA bouts was related to performance on the Stroop test. A longer average length of MVPA bouts was related to better performance on the Stroop test. When VO_2max_ was added to the model, the strength of this relationship was attenuated into a tendency. Average length of MVPA bouts was not related to performance in any other cognitive test. There was no relationship between any bout measures of SB and any cognitive test.

### 3.3. Associations Analysed Seperately for Fitter and Less Fit Office Workers

We performed additional analyses for the two cognitive tests that showed a relationship with VO_2max_. Participants with a fitness level above and below the VO_2max_ median were analysed separately. The median split was 40.05 mL·kg^−1^·min^−1^. In the multiple regression models for the Stroop test, 168 participants were in the less fit and 165 in the fitter group, whereas in the regression models for word recognition, 166 participants were in the less fit group and 164 in the fitter group. These regressions are displayed in Table 6 and Table 7.

In the separate analyses, % in MVPA and % in SB were not related to any of the cognitive outcomes in either fitness group. Using bouts of MVPA and SB as independent variables, a significant relationship was found between Stroop and the average length of MVPA bouts in participants with low fitness only. In the least fit participant group, longer MVPA bouts predicted better performance on Stroop. Additionally, in the group of participants with high fitness, the average length of MVPA bouts was related to poorer performance on word recognition.

## 4. Discussion

This study investigated possible associations between PA patterns and cognitive functions in a relatively active sample of office workers. The main findings were that the proportion of time spent in MVPA was not associated with any of the cognitive outcomes, but among the less fit half of the office workers, longer average length of MVPA bouts was related to better Stroop performance, while for the fitter individuals, longer average length of MVPA bouts was related to worse word recognition. Also, the proportion of time spent in SB tended to be positively associated to the number of words recalled in free recall. These findings suggest that MVPA and SB are not strong predictors of cognitive function in this group of active office workers and that the importance of prolonged bouts of MVPA appears to be dependent on fitness level.

### 4.1. Effects of Average Daily MVPA and SB

The findings that the proportion of time spent in MVPA was not associated with any of the cognitive outcomes was somewhat unexpected, since a recently published investigation in the same population showed that aerobic capacity was positively associated with performance in the Stroop and word recognition tests [6]. That study did, however, only consider cardiorespiratory fitness and did not investigate the importance of PA and SB.

In a recent systematic review on the cross-sectional association between PA and cognitive function in adults, Cox et al. found limited evidence for a positive relationship between PA and cognitive function [10]. They identified only one study that used objective measures of PA [27]. Boucard et al. categorised participants as active or sedentary using a combination of self-reported measures and accelerometer data. Accelerometer data was based on two days of wear time only, which has been shown to be insufficient for obtaining valid PA data in adults [21,22], and thus difficult to compare to our results. Nevertheless, Boucard et al. found no significant differences between individuals classified as active and those classified as sedentary for any of the cognitive outcomes in the young adult (age 18–28) and young old (age 60–70) groups, which is in line with the results of the present study. However, they showed that active compared to sedentary participants in the oldest participant group (age 71–81) had, significantly better inhibition, suggesting that PA is particularly important for retaining executive functions at an older age. Among middle aged adults, a recent study showed no association between MVPA and executive functions, which is in line with the results of the present study [28].

When addressing the relationship between SB and cognitive outcomes, again very few studies have used objective measures [17]. However, Rosenberg et al. found that objectively measured sedentary time was not related to any of the cognitive outcomes in older individuals [18]. Conversely, faster completion of TMT-A was related to higher self-reported SB. In the present study, we identified a tendency towards a positive association between objectively measured SB and one of the investigated cognitive functions. More time spent in SB tended to be related to a higher number of words recalled in free recall. Free recall has previously been shown to be useful in detecting dementia 10 years prior to clinical diagnosis [29]. Since self-reported physical inactivity has been shown to be the one of the largest risk factors for developing Alzheimer’s disease [30], our results are somewhat unanticipated.

Participants in the present study exhibited relatively high levels of MVPA and good fitness levels, which may serve as a buffer for long daily sitting time [13]. Our results also contrast with the results of a recent review on the association between sedentary behaviour and cognitive function. Falck et al. concluded that SB seems to be associated with poorer cognitive function yet, the vast majority of included studies used TV viewing as a proxy for sedentary activity. This may suggest that cognitive engagement during sedentary periods, which we cannot obtain with objective measures, could influence the effects of SB on cognitive function. Nevertheless, self-reported measures may under or overestimate SB depending on the nature of the question asked [31]. An explanation for the tendency towards a positive relationship between free recall and SB could be that participants with longer time in SB may engage in more mentally demanding activities, which might enhance episodic memory. Furthermore, Vásquez et al. showed a distinct relationship between SB and executive function in women compared to men. In women, more time in SB was associated with worse executive function, while for men, more time in SB was associated with better executive function [28]. While differences between genders were observed in some of the cognitive tests in our study, we did not investigate gender-specific relationships between SB and cognitive functions.

While our data showed no associations between total time in MVPA and any of the tested cognitive functions, improving CRF has previously shown a robust effect on cognitive performance in older adults [32]. However, the cross-sectional relationship between cognitive function and CRF appears weak [33]. While most of these studies have been performed on older individuals, we have recently shown that higher CRF is associated with Stroop and word recognition performance up to a VO_2max_ of 44 mL⋅kg^−1^⋅min^−1^ in a healthy working-age population [6], and CRF could potentially mediate the relationship between PA, SB and cognitive function. However, VO_2max_ did not influence the relationship on any cognitive outcome when added to the regression models. Since CRF is affected by high intensities of PA behaviour, we expected that time spent in MVPA would be associated to the same cognitive outcomes as CRF. The absence of such an association might be related to the poor ability of the ActiGraph to capture high frequency accelerations [34], which reflect high intensity activities that potentially could enhance CRF. A set cut-off point for quantifying time spent in different behaviours is useful for describing the absolute intensity of the activity, but it is not able to capture the relative intensity of PA for each individual, thus a cut-off point relative to the individual VO_2max_ may be a more precise measure [35]. In our relatively fit sample, the cut-off point used for MVPA might have been too low to capture fitness-enhancing physical activities.

### 4.2. Prolonged Bouts of MVPA and SB

The global recommendations for PA suggests that MVPA should be performed in bouts of at least 10 min for improving health [36]. This is currently highly debated because of the increasing numbers of studies showing that total time in MVPA is related to improved health and reduced mortality [37] and some national guidelines have now changed their recommendations to include all activity [38]. Almost all participants in the current study reached the recommendations of 150 min/week of total time, but considerably fewer reached 150 min/ week if only time spent in 10 min bouts were included (see Table 1). Our finding that average length of MVPA bouts was related to Stroop performance may imply that longer durations of MVPA are needed to positively influence executive functions. This is partly supported by a recent paper showing an effect on cognition from activity accumulated in bouts but not total time [39].

Prolonged SB bouts have been associated with increased cardio-metabolic risk [16]. Time spent in sedentary bouts of 10–20 min have also been associated with enhanced academic achievement in young adults [40]. However, we did not observe an association between prolonged SB bouts of at least 20 min and cognitive performance in the present sample of middle-age adults. Studies are difficult to compare, due to the diversity in populations and measurement methods used for both cognitive functions and SB. Experimental studies on the effect of breaking up prolonged SB bouts have shown mixed results. One study showed that breaking up prolonged sitting acutely decreased fatigue in overweight/obese adults, whereas cognitive performance was unaffected [41]. Another study showed that activity breaks were detrimental to executive performance [42]. Wheeler and colleagues showed that a combination of morning exercise and breaking up sitting enhanced working memory but affected executive functions negatively [43]. Thus, more research is needed to address the effect of prolonged SB on cognitive functions.

### 4.3. MVPA and SB for Different Levels of Fitness

When analysing the associations between prolonged bouts of MVPA and/or SB with Stroop and word recognition for participants with high and low fitness separately, we found that for the least fit subject group, the average length of MVPA bouts was significantly related to better performance on Stroop. Additionally, the length of MVPA bouts was related to worse performance on word recognition for the fitter subject group. These rather complex associations between physical activity patterns and cognitive functions among office workers suggests that a sub-group analysis of low fitness and high fitness individuals may be necessary to fully understand how PA and SB are related to cognitive function. Longer time in MVPA bouts will be more physiologically challenging for persons with low fitness compared to those with high fitness because the MVPA cut-off does not take fitness level into account. The results that longer bouts are associated with better performance may suggest that these long bouts could induce improvements in cardiovascular health in the low fitness participants. Thus, while this study shows no association between any of the cognitive variables and total time spent in SB or MVPA, we do show that Stroop performance, previously shown to be related to fitness, was related to the average length of MVPA bouts among the low fitness participants only. The fact that longer bouts was related to worse performance on word recognition could indicate that long bouts of MVPA may not be advantageous for high fitness individuals’ cognition.

This suggests that health promotion strategies aiming to support cognitive function may be more effective if they aim to prolong MVPA bouts specifically in low fitness individuals, rather than promoting more MVPA among all office workers. Long-term intervention studies with long follow-ups are needed to describe and interpret the effect of PA and SB on cognitive functions.

### 4.4. Limitations

It is important to acknowledge that since this is a cross-sectional study and that any associations identified or not do not substantiate causal relationships. Furthermore, we could not ascertain that other confounding factors (e.g., sleep, alcohol consumption, depression, BMI, etc.) could have influenced the results.

The subject sample was relatively healthy, with good fitness levels and a high proportion reaching the WHO PA recommendations [36]. PA patterns may be more related to cognitive function among less fit and or less active office workers. Many Swedish companies have already implemented environmental changes to improve employee’s health, well-being, and ergonomics. The two collaborating companies in the present project have progressive policies for their employees’ health, which might bias the population investigated. In Sweden, many office workers have access to a standing workstation with an adjustable desk and are provided with tax-deductible corporate wellness benefits, such as gym memberships. These factors might explain why our sample of office workers were of relatively high fitness.

Leisure time PA and occupational PA have different relations to overall health [44] and this might also be reflected in the association between PA, SB, and cognitive functions. Future investigations can therefore build upon the present investigation by studying whether the relationship between cognition and PA and SB differs between leisure and occupational time.

The battery of cognitive testing was originally developed to test an older population [26]. We observed ceiling effects for some of the tests, indicating that the battery may have been too easy, thus limiting our chances of identifying associations between PA and cognitive function in this working age sample. The combination of an active sample and a test battery insensitive to differences between participants may challenge the interpretation of the results.

## 5. Conclusions

In conclusion, our findings suggest that in physically active office workers, the association between objective measures of PA patterns and cognitive functions is weak. Interestingly, the length of MVPA bouts was related to better inhibition and to worse episodic memory in the least and most fit participants, respectively. These results may imply that the effect of MVPA bouts is dependent on individual fitness levels. This could have implications for organisations promoting employees’ physical and mental health. The present study, however, does not allow for conclusions on causality. Thus, controlled intervention studies with long follow-ups are needed to investigate how changes in MVPA and SB affect cognitive functions in office workers.

## Figures and Tables

**Table 1 ijerph-16-04721-t001:** Characteristics and cognitive performance of the participants.

Descriptive	n	Min	Max	Mean	SD
Age (year)	334	21	66	42.43	9.12
Education (year)	334	9	22	14.43	2.29
VO_2max_ (mL⋅kg^−1^⋅min^−1^) ^†^	334	18.50	62.27	39.99	8.33
% in MVPA ^‡^	334	1.70	17.22	6.76	2.42
% in Sedentary	334	34.47	76.09	59.13	7.00
Daily average time in MVPA bouts (min/day)	334	0.00	99.00	25.44	19.45
Daily average time in sedentary bouts (min/day)	334	39.40	486.30	217.82	84.91
Average length of MVPA bout (min)	334	0.00	69.50	18.49	7.08
Average length of sedentary bouts (min)	334	26.00	49.50	33.53	3.75
Female	68%				
Fulfil recommendation of 150 min/week of MVPA in bouts of 10 min	48%				
Fulfil recommendation of 150 min/week of all MVPA	99.7%				
Stroop colour and word (seconds)	333	30.32	103.40	48.13	9.52
TMT-B (seconds) ^§^	289	24.00	113.25	50.97	14.41
2–back (accuracy, Max = 80.00)	332	25.00	80.00	72.08	7.39
Digit symbol (mean ms for correct response)	333	1405.30	3389.92	2208.59	364.55
TMT-A (seconds) ^§^	332	11.19	44.66	20.65	5.92
2–back (mean milliseconds)	331	542.21	1153.06	795.49	121.33
Free recall (recalled words, Max = 16)	333	2.00	15.00	8.60	2.46
Word recognition (recalled words, Max = 30)	333	12.00	30.00	23.44	3.82
Digit span backwards (highest span achieved)	333	2.00	8.00	5.27	1.39
AOS (sum of perfectly recalled sets) ^¶^	307	3.00	49.00	19.59	10.82
AOS (accuracy, number of letters in correct position)	308	5.00	53.00	33.85	9.23

^†^ VO_2max_ = estimated maximal oxygen uptake from Ekblom–Bak cycle ergometer test. ^‡^ MVPA = moderate-to-vigorous physical activity. ^§^ TMT-B = trail-making task B. TMT-A = trail-making task A. ^¶^ AOS = automated operation span.

**Table 2 ijerph-16-04721-t002:** Multiple linear regressions for executive functions and processing speed—cognitive functions were used as dependent variables.

Cognitive function	Model 1 ^†^	Model 2 ^‡^	Model 3 ^§^		Model 4 ^¶^	
	Log % in MVPA	% in SB	Log % in MVPA	VO_2max_	% in SB	VO_2max_
**Executive functions**	β (99% CI)	β (99% CI)	β (99% CI)	β (99% CI)	β (99% CI)	β (99% CI)
Stroop (2xlog)	−0.011 (−0.152 to 0.130)	0.012 (−0.131 to 0.155)	0.012 (−0.130 to 0.155)	−0.159 * (−0.332 to 0.013)	−0.016 (−0.162 to 0.129)	−0.161 * (−0.334 to 0.013)
TMT-B	0.034 (−0.116 to 0.182)	−0.032 (−0.194 to 0.126)	0.044 (−0.109 to 0.196)	−0.065 (−0.260 to 0.130)	−0.044 (−0.210 to 0.118)	−0.065 (−0.261 to 0.130)
2–back	−0.001 (−0.141 to 0.140)	0.053 (−0.088 to 0.197)	−0.004 (−0.147 to 0.139)	0.022 (−0.150 to 0.195)	0.060 (−0.085 to 0.207)	0.037 (−0.137 to 0.210)
**Processing speed**						
Digit symbol	0.001 (−0.138 to 0.141)	−0.004 (−0.145 to 0.138)	0.008 (−0.134 to 0.150)	−0.045 (−0.216 to 0.126)	−0.012 (−0.157 to 0.133)	−0.047 (−0.219 to 0.126)
TMT-A (2xlog)	0.013 (−0.130 to 0.156)	0.042 (−0.103 to 0.190)	0.017 (−0.129 to 0.162)	−0.025 (−0.202 to 0.152)	0.040 (−0.108 to 0.191)	−0.012 (−0.190 to 0.166)
2–back RT (log)	0.075 (−0.064 to 0.214)	−0.053 (−0.195 to 0.087)	0.074 (−0.067 to 0.216)	0.004 (−0.168 to 0.175)	−0.052 (−0.198 to 0.092)	0.006 (−0.167 to 0.179)

All models are adjusted for age, gender and education. ^†^ Model 1—adjusted for % of daytime in moderate-to vigorous physical activity (% in MVPA) logged to the base of 10. ^‡^ Model 2—adjusted for % of daytime in sedentary behaviour (% in SB). ^§^ Model 3—as model 1 but additionally adjusted for VO_2max_. ^¶^ Model 4—as model 2 but additionally adjusted for VO_2max_. Standardised beta coefficients (β) and 99% confidence intervals (99% CI) are given for every independent variable in the model. Statistically significant beta coefficients with a corresponding *p*-value < 0.01 are marked with **. Beta coefficients with 0.01 < *p* < 0.05 are treated as a tendency and are marked with *. TMT-B = trail making task B. TMT-A = trail making task A. The 2–back RT = 2–back response time.

**Table 3 ijerph-16-04721-t003:** Multiple linear regressions for episodic memory and working memory—cognitive functions were used as dependent variables.

Cognitive Function	Model 1 ^†^	Model 2 ^‡^	Model 3 ^§^		Model 4 ^¶^	
	Log % in MVPA	% in SB	Log % in MVPA	VO_2max_	% in SB	VO_2max_
**Working memory**	β (99% CI)	β (99% CI)	β (99% CI)	β (99% CI)	β (99% CI)	β (99% CI)
Digit span backwards	−0.006 (−0.146 to 0.133)	0.003 (−0.139 to 0.146)	−0.001 (−0.143 to 0.141)	−0.037 (−0.209 to 0.136)	−0.003 (−0.149 to 0.142)	−0.038 (−0.211 to 0.136)
AOS recalled sets	0.062 (−0.080 to 0.206)	−0.011 (−0.159 to 0.136)	0.046 (−0.098 to 0.191)	0.110 (−0.064 to 0.280)	0.010 (−0.140 to 0.161)	0.122 (−0.052 to 0.293)
AOS accuracy	0.064 (−0.074 to 0.203)	−0.022 (−0.165 to 0.120)	0.052 (−0.089 to 0.194)	0.079 (−0.090 to 0.244)	−0.006 (−0.152 to 0.140)	0.089 (−0.081 to 0.256)
**Episodic memory**						
Free recall	−0.017 (−0.154 to 0.121)	0.125 * (−0.011 to 0.266)	−0.021 (−0.161 to 0.118)	0.031 (−0.138 to 0.199)	0.136 * (−0.003 to 0.280)	0.061 (−0.107 to 0.229)
Word recognition	0.045 (−0.095 to 0.185)	0.035 (−0.107 to 0.177)	0.018 (−0.123 to 0.158)	0.187 ** (0.017 to 0.358)	0.071 (−0.071 to 0.215)	0.209 ** (0.038 to 0.380)

All models are adjusted for age, gender, and education. ^†^ Model 1–adjusted for % of daytime in moderate-to-vigorous physical activity (% in MVPA) logged to the base of 10. ^‡^ Model 2–adjusted for % of daytime in sedentary behaviour (% in SB). ^§^ Model 3–as model 1 but additionally adjusted for VO_2max_. ^¶^ Model 4–as model 2 but additionally adjusted for VO_2max_. Standardised beta coefficients (β) and 99% confidence intervals (99% CI) are given for every independent variable in the model. Statistically significant beta coefficients with a corresponding *p*-value < 0.01 are marked with **. Beta coefficients with 0.01 *< p <* 0.05 are treated as a tendency and are marked with *. AOS = automated operation span.

**Table 4 ijerph-16-04721-t004:** Multiple linear regressions with detailed measurements of physical activity—cognitive functions were used as dependent variables.

Cognitive function	Model 1 ^†^	Model 2 ^‡^	Model 3 ^§^	Model 4 ^¶^
	Daily avg MVPA bouts	Length of MVPA bouts	Daily avg MVPA bouts	Length of MVPA bouts
**Executive functions**	β (99% CI)	β (99% CI)	β (99% CI)	β (99% CI)
Stroop (2xlog)	−0.078 (−0.216 to 0.061)	−0.155 ** (−0.293 to −0.017)	−0.049 (−0.191 to 0.094)	−0.134 * (−0.275 to 0.006)
TMT-B (log)	0.000 (−0.148 to 0.148)	−0.077 (−0.228 to 0.075)	0.012 (−0.142 to 0.165)	−0.070 (−0.227 to 0.086)
2–back accuracy	−0.001 (−0.139 to 0.137)	0.102 (−0.036 to 0.241)	−0.006 (−0.149 to 0.137)	0.103 (−0.038 to 0.245)
**Processing speed**				
Digit symbol	0.028 (−0.109 to 0.165)	−0.068 (−0.206 to 0.069)	0.040 (−0.102 to 0.181)	−0.063 (−0.204 to 0.077)
TMT-A (2xlog)	0.036 (−0.105 to 0.176)	−0.092 (−0.235 to 0.049)	0.042 (−0.103 to 0.187)	−0.092 (−0.238 to 0.052)
2–back RT (log)	0.109 (−0.028 to 0.245)	−0.048 (−0.186 to 0.090)	0.112 (−0.030 to 0.252)	−0.054 (−0.195 to 0.087)
**Working memory**				
Digit span backwards	−0.003 (−0.140 to 0.134)	0.019 (−0.120 to 0.157)	0.005 (−0.137 to 0.147)	0.026 (−0.115 to 0.168)
AOS recalled sets	0.080 (−0.060 to 0.218)	0.038 (−0.104 to 0.182)	0.058 (−0.086 to 0.201)	0.017 (−0.129 to 0.163)
AOS accuracy	0.067 (−0.069 to 0.202)	−0.016 (−0.155 to 0.123)	0.051 (−0.090 to 0.191)	−0.035 (−0.177 to 0.107)
**Episodic memory**				
Free recall	−0.010 (−0.145 to 0.125)	0.019 (−0.117 to 0.155)	−0.016 (−0.156 to 0.123)	0.015 (−0.124 to 0.154)
Word recognition	0.059 (−0.079 to 0.197)	−0.058 (−0.196 to 0.081)	0.022 (−0.119 to 0.162)	−0.095 (−0.234 to 0.045)

All models are adjusted for age, gender, education, and wear time. ^†^ Model 1–adjusted for daily average moderate-to-vigorous physical activity (MVPA). ^‡^ Model 2–adjusted for average length of MVPA bouts ^§^ Model 3–adjusted for daily average MVPA and for VO_2max_. ^¶^ Model 4–adjusted for average length of MVPA bouts and for VO_2max_. Standardised beta coefficients (β) and 99% confidence interval (99% CI) are given for every independent variable in the model. Beta coefficients with 0.01 *< p <* 0.05 are treated as a tendency and are marked with *. TMT-B = trail making task B. TMT-A = trail making task A. The 2–back RT = 2–back response time. AOS = automated operation span.

**Table 5 ijerph-16-04721-t005:** Multiple linear regressions with detailed measurements of sedentary behaviour—cognitive functions were used as dependent variables.

Cognitive function	Model 1 ^†^	Model 2 ^‡^	Model 3 ^§^	Model 4 ^¶^
	Daily avg SB bouts	Length of SB bouts	Daily avg SB bouts	Length of SB bouts
**Executive functions**	β (99% CI)	β (99% CI)	β (99% CI)	β (99% CI)
Stroop (2xlog)	−0.005 (−0.149 to 0.138)	0.066 (−0.076 to 0.209)	−0.022 (−0.166 to 0.121)	0.061 (−0.079 to 0.204)
TMT-B (log)	0.008 (−0.148 to 0.164)	0.107 (−0.045 to 0.261)	0.003 (−0.155 to 0.160)	0.106 (−0.047 to 0.260)
2–back accuracy	−0.001 (−0.143 to 0.142)	−0.034 (−0.177 to 0.107)	0.001 (−0.142 to 0.145)	−0.034 (−0.176 to 0.108)
**Processing speed**				
Digit symbol	−0.015 (−0.156 to 0.126)	0.020 (−0.121 to 0.161)	−0.019 (−0.162 to 0.122)	0.018 (−0.122 to 0.160)
TMT-A (2xlog)	0.071 (−0.072 to 0.217)	0.067 (−0.076 to 0.212)	0.070 (−0.074 to 0.217)	0.067 (−0.076 to 0.212)
2–back RT (log)	0.020 (−0.121 to 0.162)	0.055 (−0.085 to 0.196)	0.023 (−0.119 to 0.166)	0.056 (−0.084 to 0.197)
**Working memory**				
Digit span backwards	0.003 (−0.139 to 0.146)	0.018 (−0.123 to 0.160)	−0.001 (−0.144 to 0.143)	0.017 (−0.125 to 0.159)
AOS recalled sets	−0.074 (−0.219 to 0.068)	−0.036 (−0.180 to 0.107)	−0.063 (−0.208 to 0.080)	−0.033 (−0.177 to 0.110)
AOS accuracy	−0.079 (−0.220 to 0.059)	−0.060 (−0.199 to 0.078)	−0.070 (−0.212 to 0.069)	−0.057 (−0.195 to 0.081)
**Episodic memory**				
Free recall	0.095 (−0.040 to 0.236)	0.036 (−0.102 to 0.175)	0.100 (−0.037 to 0.242)	0.037 (−0.101 to 0.176)
Word recognition	0.046 (−0.095 to 0.189)	0.030 (−0.111 to 0.172)	0.067 (−0.073 to 0.210)	0.036 (−0.104 to 0.176)

All models are adjusted for age, gender, education, and wear time. ^†^ Model 1–adjusted for daily average sedentary bouts. ^‡^ Model 2–adjusted for average length of sedentary bouts. ^§^ Model 3–adjusted for daily average sedentary bouts and for VO_2max_. ^¶^ Model 4–adjusted for average length of sedentary bouts and for VO_2max_. Standardised beta coefficients (β) and 99% confidence interval (99% CI) are given for every independent variable in the model. TMT-B = trail making task B. TMT-A = trail making task A. The 2–back RT = 2–back response time. AOS = automated operation span.

**Table 6 ijerph-16-04721-t006:** Multiple linear regressions for participants with low fitness and high fitness—cognitive functions were used as dependent variables.

Split on VO_2max_	Low vs High	Model 1 ^†^	Model 2 ^‡^
		Log % in MVPA β (99% CI)	% in SB β (99% CI)
Stroop (2xlog)	Low fitness	−0.024 (−0.229 to 0.181)	0.021 (−0.177 to 0.217)
	High fitness	0.037 (−0.156 to 0.230)	−0.011 (−0.230 to 0.205)
Word recognition	Low fitness	0.067 (−0.137 to 0.273)	−0.004 (−0.201 to 0.194)
	High fitness	0.003 (−0.189 to 0.195)	0.113 (−0.096 to 0.336)

Models are adjusted for age, gender and education. ^†^ Model 1–adjusted for % of daytime in moderate-to-vigorous physical activity (% in MVPA) logged to the base of 10. ^‡^ Model 2–adjusted for % of daytime in sedentary behaviour (% in SB). Standardised beta coefficients (β) and 99% confidence interval (99% CI) are given for every independent variable in the model.

**Table 7 ijerph-16-04721-t007:** Multiple linear regressions for participants with low fitness and high fitness, with detailed measurement of physical activity and sedentary behaviour cognitive functions, were used as dependent variables.

Split on VO_2max_		Model 1 ^†^	Model 2 ^‡^	Model 3 ^§^	Model 4 ^¶^
Cognitive function	Low vs High	Daily avg MVPA bouts	Daily avg SB bouts	Length of MVPA bouts	Length of SB bouts
		β (99% CI)	β (99% CI)	β (99% CI)	β (99% CI)
Stroop (2xlog)	Low Fitness	−0.071 (−0.296 to 0.145)	0.032 (−0.166 to 0.226)	−0.211 ** (−0.494 to −0.013)	0.134 (−0.069 to 0.316)
	High Fitness	−0.029 (−0.204 to 0.150)	−0.046 (−0.275 to 0.169)	−0.079 (−0.236 to 0.097)	0.006 (−0.210 to 0.222)
Word recognition	Low Fitness	0.025 (−0.194 to 0.249)	0.034 (−0.163 to 0.229)	0.056 (−0.177 to 0.317)	−0.007 (−0.201 to 0.188)
	High Fitness	0.054 (−0.129 to 0.223)	0.075 (−0.139 to 0.304)	−0.216 ** (−0.340 to −0.016)	0.096 (−0.113 to 0.317)

Models are adjusted for age, gender, education, and wear time. ^†^ Model 1–adjusted for daily average time in moderate to vigorous physical activity (MVPA) bouts. ^‡^ Model 2–adjusted for daily average time in sedentary behaviour (SB). ^§^ Model 3–adjusted for average length of MVPA bouts. ^¶^ Model 4–adjusted for average length of SB bouts. Standardised beta coefficients (β) and 99% confidence interval (99% CI) are given for every independent variable in the model. Statistically significant beta coefficients with a corresponding *p*-value < 0.01 are marked with **.

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
