# Peer review of "Relationships between Physical Activity, Sedentary Behaviour and Cognitive Functions in Office Workers"

_ijerph, 2019, doi:10.3390/ijerph16234721_

Round 1
Reviewer 1 Report
Extremely clear, concise, and relevant paper. Some comments:
Readers without expertise in PA modalities will appreciate what some examples (walking, stairs, jogging) count in which buckets of your defined energy expenditure buckets.
Procedure is a bit unclear regarding timing of the activity monitoring. Was this always collected during the 8 days following the cognitive testing? Are there limitations to the interpretation of the results if this is the case?
Line 372: This section title mentions bouts of SB but does discuss this. This would be a particularly relevant topic to include discussion of as it relates to the analysis, as the paper is framed in the office environment and longer bouts of uninterrupted SB may be an indicator of different health risks.
I was surprised to not see any analysis of how office vs outside of the office PA might be associated with outcomes. Was this analysis already performed or out of the scope of this research? If the former, some discussion of these findings would be very relevant given the context of the paper and the title. If the latter, some future directions should be added to investigate this further. Such analysis is important as it starts to uncover associations between when and where PA interventions and programs can show the most benefit, and if very poor office habits for SB might be associated with outcomes.
Author Response
Response to Reviewer 1 Comments
Comment: Extremely clear, concise, and relevant paper. Some comments:
Response: Thank you for some really relevant comments. We have tried to address them all.
We discovered a minor mistake in the physical activity analysis and have now corrected it. All tables are therefore updated with new values. The new analysis did not influence the interpretation of the results or the conclusion.
Line specifications are given with “Track changes” disabled.
Point 1: Readers without expertise in PA modalities will appreciate what some examples (walking, stairs, jogging) count in which buckets of your defined energy expenditure buckets.

Response 1: We have added the categorization of intensities in the introductions and further added some example of PA modalities, see lines 41-43.
“The intensity of PA is often categorised into low (LIPA, including activities such as slow walking), moderate (MPA, e.g. fast walking), and vigorous (VPA, e.g. climbing stairs or running).”
Point 2: Procedure is a bit unclear regarding timing of the activity monitoring. Was this always collected during the 8 days following the cognitive testing? Are there limitations to the interpretation of the results if this is the case?
Response 2: In the last paragraph of “procedures” we have now specified that the measurement period was after the cognitive testing (Line 100).
“Activity measurement was then obtained over the subsequent eight days.”
We wanted the cognitive testing and the activity measurement to be close to each other. And we placed the activity measurement after so that we did not affect the cognitive testing.
While it cannot be excluded that the participants of this study unintentionally adapted their physical activity pattern during the measurement period, we think that the timing of the cognitive testing in relation to the physical activity measurements is unlikely to have affected the measured patterns.
Point 3: Line 372: This section title mentions bouts of SB but does discuss this. This would be a particularly relevant topic to include discussion of as it relates to the analysis, as the paper is framed in the office environment and longer bouts of uninterrupted SB may be an indicator of different health risks.
Response 3: Thank you for pointing that out. A paragraph in the discussion has now been added discussing, how different types of prolonged SB may affect cognition. Lines 388-400. From the text:
“Prolonged SB bouts have been associated with increased cardio-metabolic risk [16]. Time spent in sedentary bouts of 10-20 minutes have also been associated with enhanced academic achievement in young adults [40]. However, we did not observe an association between prolonged SB bouts of at least 20 minutes and cognitive performance in the present sample of middle-age adults. Studies are difficult to compare, due to the diversity in populations and measurement methods used for both cognitive functions and SB. Experimental studies on the effect of breaking up prolonged SB bouts have shown mixed results. One study showed that breaking up prolonged sitting acutely decreased fatigue in overweight/obese adults, whereas cognitive performance was unaffected [41]. Another study, showed that activity breaks was detrimental to executive performance [42]. Wheeler and colleagues showed that a combination of morning exercise and breaking up sitting enhanced working memory but affected executive functions negatively [43]. Thus, more research is needed to address the effect of prolonged SB on cognitive functions.”
Point 4: I was surprised to not see any analysis of how office vs outside of the office PA might be associated with outcomes. Was this analysis already performed or out of the scope of this research? If the former, some discussion of these findings would be very relevant given the context of the paper and the title. If the latter, some future directions should be added to investigate this further. Such analysis is important as it starts to uncover associations between when and where PA interventions and programs can show the most benefit, and if very poor office habits for SB might be associated with outcomes.
Response 4: Thank you for making a really good point. We have added this to limitations and suggest that this needs to be investigated further, see lines 436-439:
“Leisure time PA and occupational PA have different relations to overall health [44] and this might also be reflected in the association between PA, SB, and cognitive functions. Future investigations can therefore build upon the present investigation by studying whether the relationship between cognition and PA and SB differs between leisure and occupational time.”
Reviewer 2 Report
Thanks to the authors of the study that was aimed to explore the association between physical activity and cognitive functions among office workers in Sweden. The extant research, the gap of the study, and the research itself was well articulated. There are only a few specific and general comments to consider as below:
Introduction
Line 43: Please define the maximal oxygen uptake (VO2max) in its first appearance. The authors should not assume that all the readers know VO2max.
The essence of the study must be clearly stated.
Methods
The reason for selecting only two Swedish companies needs to be given. How many companies are in Sweden and why only these two? What is the nature of their operation?
Line 105: Punctuation?
Line 119: It is the first time meeting SEB without defining this abbreviation.
Line 123-127: Did you perform a reliability analysis of the cognitive test? If not, it would be of value if the authors included the reliability scores in the manuscript.
Results
Line 220: Be clear that it is “log % in MVPA…” which is different from “% in MVPA”
Spell out, in the footnote, all the abbreviations used in the Tables e.g. AOS, MVPA, SB (Table 1) etc. Also in other Tables.
Discussion
Please be careful when interpreting the log-transformed values. One should not interpret log-transformed values as if they were actual values.
What is the policy implication of the findings?
Limitations
What are the biases associated with using wearable devices in relation to non-supervision of participants, specificity and sensitivity?
The limitation of using log-transformed values need to be indicated.
Conclusion
The conclusion needs to answer the objective of the study, yet it deviates from the objective.
Author Response
Response to Reviewer 2 Comments
Comment: Thanks to the authors of the study that was aimed to explore the association between physical activity and cognitive functions among office workers in Sweden. The extant research, the gap of the study, and the research itself was well articulated. There are only a few specific and general comments to consider as below:
Response: Thank you for your feedback. You have made several good points that we have tried to address.
We discovered a minor mistake in the physical activity analysis and have now corrected it. All tables are therefore updated with new values. The new analysis did not influence the interpretation of the results or the conclusion.
Line specifications are given with “Track changes” disabled.
Point 1: Introduction
Line 43: Please define the maximal oxygen uptake (VO2max) in its first appearance. The authors should not assume that all the readers know VO2max. The essence of the study must be clearly stated.
Response 1:
Thank you for pointing that out. We have now defined maximal oxygen consumption (Line 47) Thank you for making that point. We have now rewritten the sentence including the aim of this study (lines 75-80). We hope that essence became clearer to the reader. From the text:
“To properly address the difficulties in using self-reported measurement of PA [11] and SB [19] we aimed to investigate if and to what degree objective measures of PA and SB are related to cognitive functions among office workers. We hypothesized that more average daily time spent in moderate-to-vigorous PA (MVPA) and less time spent in SB would be related to better cognitive task performance independently of CRF and that the associations would be even stronger for prolonged bouts of PA and SB in lower fit individuals.”
Point 2: Method:
The reason for selecting only two Swedish companies needs to be given. How many companies are in Sweden and why only these two? What is the nature of their operation?
Response 2: This study is funded by the a funding body named ”The knowledge foundation”. The knowledge foundation facilitates strategic collaborations between the academia and the private sector. These two companies were interested in such collaboration. We have highlighted this in the materials and methods paragraph (see line 83-85):
“In a cross-sectional design, 1940 office workers from two Swedish companies (Intrum and ICA-gruppen) were invited to participate in the study. These companies were part of a research collaboration to coproduce knowledge in a series of studies on how PA and SB relate to healthy brain functions.”
Furthermore, we have addressed the limitations of investigating such population in the limitation part (Line 427-435).
“The subject sample was relatively healthy, with good fitness levels, and a high proportion reaching the WHO PA recommendations [36]. PA patterns may be more related to cognitive function among less fit and or less active office workers. Many Swedish companies have already implemented environmental changes to improve employee’s health, well-being, and ergonomics. The two collaborating companies in the present project have progressive policies for their employees’ health, which might bias the population investigated. In Sweden, many office workers have access to a standing workstation with an adjustable desk and are provided with tax-deductible corporate wellness benefits, such as gym memberships. These factors might explain why our sample of office workers were of relatively high fitness.”
Point 3:
Line 105: Punctuation? Line 119: It is the first time meeting SEB without defining this abbreviation. Line 123-127: Did you perform a reliability analysis of the cognitive test? If not, it would be of value if the authors included the reliability scores in the manuscript.
Response 3:
Thank you. This is now corrected. Thanks for noticing this mistake. We have now replaced SEB with SB. Thank you for pointing that out. This is extremely important. In the study of Jonasson 2017 they showed that all tests had good reliability and validity. We have now added a sentence to the method in the paragraph describing the neuropsychological test battery, see line 129-131:
“The battery was similar to Jonasson et al. 2017 [26] with slight adaptations made to the Trail Making Test A and B. All tests showed good validity and reliability for measurement of the respective cognitive domains [26].”
Point 4: Results
Line 220: Be clear that it is “log % in MVPA…” which is different from “% in MVPA” Spell out, in the footnote, all the abbreviations used in the Tables e.g. AOS, MVPA, SB (Table 1) etc. Also in other Tables.
Response 4:
This is now changed accordingly. Footnotes are now spelled out in all tables.
Point 5: Discussion
Please be careful when interpreting the log-transformed values. One should not interpret log-transformed values as if they were actual values. What is the policy implication of the findings?
Response 5:
Thank you for pointing that out. It is definitely problematic to interpret log-values, but as we are using standardised beta coefficients and discuss these results and not the actual log transformed values we prefer to present data as such. One policy implication of these findings are that strategies to promote cognitive functions through physical activity should include interventions that focuses at increasing cardiorespiratory fitness among low fit individuals. This is now clarified on lines 418-421.
“This suggests that health promotion strategies aiming to support cognitive function may be more effective if they aim to prolong MVPA bouts specifically in low fitness individuals, rather than promoting more MVPA among all office workers. Long-term intervention studies with long follow-ups are needed to describe and interpret the effect of PA and SB on cognitive functions.”
Point 6: Limitations
What are the biases associated with using wearable devices in relation to non-supervision of participants, specificity and sensitivity? The limitation of using log-transformed values need to be indicated.
Response 6:
Of course non-supervision of the subjects is a limitation of measuring physical activity with accelerometers. We used a combination of a software (actilife) wear time validation (0 counts for 60 minutes addresses as non-wear time) and activity diaries to address this problem. The wear time validation process identifies time periods where the monitor was not worn and exclude this from the analysis. And the diaries should complement this, by adding sleep periods. The bias could be that we have longer periods of SB where the non-wear time did not kick in, however this is unlikely to affect the results. In addition, supervision of a large populations would not be feasible. As stated above, our understanding is that the use of standardized beta values circumvents the limitations in applying log-values.
Point 7: Conclusion
The conclusion needs to answer the objective of the study, yet it deviates from the objective.
Response 7:
We have now corrected the objectives (lines 75-80)
“To properly address the difficulties in using self-reported measurement of PA [11] and SB [19] we aimed to investigate if and to what degree objective measures of PA and SB are related to cognitive functions among office workers. We hypothesized that more average daily time spent in moderate-to-vigorous PA (MVPA) and less time spent in SB would be related to better cognitive task performance independently of CRF and that the associations would be even stronger for prolonged bouts of PA and SB in lower fit individuals.”
… and the conclusion (lines 447-450) so that the conclusion is better aligned with the objective:
“In conclusion, our findings suggest that in physically active office workers the association between objective measures of PA patterns and cognitive functions is weak. Interestingly, the length of MVPA bouts was related to better inhibition and to worse episodic memory in the least and most fit participants, respectively.”
Reviewer 3 Report
Comments to the Authors
Relationships between Physical Activity, Sedentary Behaviour and Cognitive Functions in Office Workers
Text:
While the authors did a good job at justifying why their research question is worthwhile, I am hesitant to believe that these results are valid. I have several concerns:
Residual confounding. There are other covariates that could be associated with PA, SB and cognitive function that are not controlled for in the models. Sleep duration could be one such variable; OH drinking; BMI; comorbidities.Why did the authors not adjust for PA and SB in the same models?
Given the many tests conducted, a P value <0.01 will not be close to taking that into account; which begs the questions whether the significant results obtained were truly valid.
Minor comments:
A few grammatical errors need to be corrected.
Overall level of enthusiasm:
Overall, the topic is very interesting. However, I have concerns about the validity of these results.
Additional comments:
None.
Author Response
Response to Reviewer 3 Comments
Thank you for some really relevant comments. We have tried to address them all.
We discovered a minor mistake in the physical activity analysis and have now corrected it. All tables are therefore updated with new values. The new analysis did not influence the interpretation of the results or the conclusion, but the association between sedentary behaviour and free recall became a tendency. We have therefore clarified in the results and discussion that the association was weak.
Line specifications are given with “Track changes” disabled.
Point 1: While the authors did a good job at justifying why their research question is worthwhile, I am hesitant to believe that these results are valid. I have several concerns:
Residual confounding. There are other covariates that could be associated with PA, SB and cognitive function that are not controlled for in the models. Sleep duration could be one such variable; OH drinking; BMI; comorbidities.
Response 1: This is a really good point. Models could have been adjusted for several variables. We selected the variables that we know have a strong influence on cognitive function such as age, education and gender, but models could have been adjusted for more covariates. Another approach could be a mediation analysis to investigate which variables that mediate this complex relationship, but this is out of the scope for this particular article and would be more suitable in longitudinal designs. We added that there may be residual confounding factors in the limitation section in the discussion, see lines 423-426:
“It is important to acknowledge that since this is a cross-sectional study, any associations identified or not, do not substantiate causal relationships. Furthermore, we could not ascertain that other confounding factors (e.g. sleep, alcohol consumption, depression, BMI etc.) could have influenced the results.”
Point 2: Why did the authors not adjust for PA and SB in the same models?
Response 2: % in PA and % in SB were highly correlated and thus violated the assumption of collinearity in linear regression analysis. Thus, a compositional data analysis, treating data as sub-compositions of a whole is the next logical step. We are currently learning the methodology of compositional data analysis to address the problem of collinearity in activity measurements for future studies.
Point 3: Given the many tests conducted, a P value <0.01 will not be close to taking that into account; which begs the questions whether the significant results obtained were truly valid.
Response 3: One strength of this study is the inclusion of many cognitive tests. We agree that when investigating many outcome variables there is a risk of making type I errors, but correcting even more might also make us susceptible to type II errors. We have now clarified better in the conclusion that in this physically active population of office workers, MVPA and SB appear to be only very weakly associated to cognitive functions, see lines 447-448.
“In conclusion, our findings suggest that in physically active office workers the association between objective measures of PA patterns and cognitive functions is weak.”
Point 4: Minor comments: A few grammatical errors need to be corrected.
Response 4: We have attempted to identify and correct these.
Point 5: Overall level of enthusiasm:
Overall, the topic is very interesting. However, I have concerns about the validity of these results.
Response 5: We have made an honest attempt to respond to your raised concerns and to make amendments to the manuscript to address each of the concerns.
Reviewer 4 Report
This article provides sufficient background data and statistical analysis, and overall is well-structured. However, here are some suggestions for your reference:
There have been similar studies, so please clearly explain the main contribution and novelty of this research at the very beginning. At the end, please clearly strengthen the parts of "suggestion", "implication" and so on. Please re-check the layout and article structure (especially paragraph 1st). In addition, in general, for a formal journal, it is not appropriate that a paragraph contains only one sentence (e.g., Line73, Line 244-245, …) Format in some paragraphs look a bit messy (2.2.3) and should be improved, such as Line134 (Episodic memory), Line135 (Free Recall), Line145 (Digit symbol) ... . In Method, please present the refusal rate of sampling, and explains the possible impact, if any, on the experiment. Please try to briefly describe important terms/abbreviations in the body, for example: : PHIBRA (Line 125). Please re-check the statistical symbols. For example, P (e.g., Line 204) should be presented in italics.Author Response
Response to Reviewer 4 Comments
Comment: This article provides sufficient background data and statistical analysis, and overall is well-structured. However, here are some suggestions for your reference:
Response: Thank you for good constructive feedback of this article.
We discovered a minor mistake in the physical activity analysis and have now corrected it. All tables are therefore updated with new values. The new analysis did not influence the interpretation of the results or the conclusion.
Line specifications are given with “Track changes” disabled.
Point 1: There have been similar studies, so please clearly explain the main contribution and novelty of this research at the very beginning.
Response 1: Thank you for making a good point. We have now rewritten some parts in the introduction so that it is clearer for the reader. See lines 75-80:
“To properly address the difficulties in using self-reported measurement of PA [11] and SB [19] we aimed to investigate if and to what degree objective measures of PA and SB are related to cognitive functions among office workers. We hypothesized that more average daily time spent in moderate-to-vigorous PA (MVPA) and less time spent in SB would be related to better cognitive task performance independently of CRF and that the associations would be even stronger for prolonged bouts of PA and SB in lower fit individuals.”
Point 2: At the end, please clearly strengthen the parts of "suggestion", "implication" and so on.
Response 2: We have highlighted the implications in both the discussion and conclusion. (Lines 418-421 and 450-454)
Lines 418-421:
“This suggests that health promotion strategies aiming to support cognitive function may be more effective if they aim to prolong MVPA bouts specifically in low fitness individuals, rather than promoting more MVPA among all office workers. Long-term intervention studies with long follow-ups are needed to describe and interpret the effect of PA and SB on cognitive functions.”
Lines 450-454:
“These results may imply that the effect of MVPA bouts is dependent on individual fitness levels. This could have implications for organisations promoting employees’ physical and mental health. The present study, however, does not allow for conclusions on causality. Thus, controlled intervention studies with long follow-ups are needed to investigate how changes in MVPA and SB affect cognitive functions in office workers.”
Point 3: Please re-check the layout and article structure (especially paragraph 1st). In addition, in general, for a formal journal, it is not appropriate that a paragraph contains only one sentence (e.g., Line73, Line 244-245, …) Format in some paragraphs look a bit messy (2.2.3) and should be improved, such as Line134 (Episodic memory), Line135 (Free Recall), Line145 (Digit symbol) ... .
Response 3: Thank you for pointing that out. We have tried to improve the layout according to your suggestions.
Point 4: In Method, please present the refusal rate of sampling, and explains the possible impact, if any, on the experiment.
Response 4: Thank you for making a really good point. We have now added the participation in percentage we had in the study to the first paragraph of materials and methods, see line 82-86:
“Participants/study population (See table 1a for demographics). In a cross-sectional design, 1940 office workers from two Swedish companies (Intrum and ICA-gruppen) were invited to participate in the study. These companies were part of a research collaboration to coproduce knowledge in a series of studies on how PA and SB relate to healthy brain functions. Of the 1940 invited employees, 547 (28 %) answered the questionnaire and 369 (19%) participants took part in all measurements.”
Point 5: Please try to briefly describe important terms/abbreviations in the body, for example: PHIBRA (Line 125).
Response 5: PHIBRA was the name of the study but we have now changed it to the reference name. Line 129
Point 6: Please re-check the statistical symbols. For example, P (e.g., Line 204) should be presented in italics.
Response 6: Thank you for noticing this. We have now checked the statistical symbols and made the p-values in italic.
Round 2
Reviewer 3 Report
Point 2: Why did the authors not adjust for PA and SB in the same models?
Response 2: % in PA and % in SB were highly correlated and thus violated the assumption of collinearity in linear regression analysis. Thus, a compositional data analysis, treating data as sub-compositions of a whole is the next logical step. We are currently learning the methodology of compositional data analysis to address the problem of collinearity in activity measurements for future studies.
Comment on response 2: I do not necessarily agree that these 2 constructs cannot be in the same model. In fact, studies have shown that these 2 constructs could have independent effects on the outcome. “Although some have suggested that a correlation coefficient >0.6 should be carefully considered when including in a model, others view the removal of confounding as more important, and that collinear variables of up to 0.90 can still be reasonably controlled in various models (Pischon, T, 2005).”
Author Response
Comment on response 2: I do not necessarily agree that these 2 constructs cannot be in the same model. In fact, studies have shown that these 2 constructs could have independent effects on the outcome. “Although some have suggested that a correlation coefficient >0.6 should be carefully considered when including in a model, others view the removal of confounding as more important, and that collinear variables of up to 0.90 can still be reasonably controlled in various models (Pischon, T, 2005).”"
Respone:
Thank you for the proposal. Sedentary behaviour and physical activity are interrelated rather than independent associated with for example all-cause mortality (Ekelund, 2016, Lancet, BMJ 2019). Therefore, adding both to the same model may complicate the interpretation. Thus, analysing participants with high and low sedentary behaviour separately (stratified analysis) may be advantageous. As we now reran the analyses separately for participants with high and low amounts of sedentary behaviour, moderate to vigorous physical activity was still not related to any cognitive outcome. Since the new analysis did not change the interpretation of the results, we prefer reporting the results as already presented.
Reviewer 4 Report
This article is well-written and ready for published. Here are some suggestions for authors:
1. Please recheck the reporting statistics of your manuscript based on the writing style of
IJERPH. Please especially pay attention to issues of italics, spacing, and
uppercase/lowercase.
2. Recheck table 6th and 7th about the fonts and lines (thickness, dashed line, etc.).
3. Please add nationality or more other information about the grant awarded of this research.
Author Response
This article is well-written and ready for published. Here are some suggestions for authors:
1. Please recheck the reporting statistics of your manuscript based on the writing style of
IJERPH. Please especially pay attention to issues of italics, spacing, and
uppercase/lowercase.
2. Recheck table 6th and 7th about the fonts and lines (thickness, dashed line, etc.).
3. Please add nationality or more other information about the grant awarded of this research.
Response:
Thank you. We have now rechecked the writing style and changed this accordingly We have now checked the tables and updated the fronts and lines. We have now added the nationality of the research grant.